# Noncured Graphene Thermal Interface Materials for High-Power Electronics: Minimizing the Thermal Contact Resistance

**DOI:** 10.3390/nano11071699

**Published:** 2021-06-28

**Authors:** Sriharsha Sudhindra, Fariborz Kargar, Alexander A. Balandin

**Affiliations:** Phonon Optimized Engineered Materials Center, Department of Electrical and Computer Engineering, University of California, Riverside, CA 92521, USA

**Keywords:** surface roughness, thermal contact resistance, thermal conductivity, graphene, silicone oil, thermal interface materials

## Abstract

We report on experimental investigation of thermal contact resistance, $R_{C}$, of the noncuring graphene thermal interface materials with the surfaces characterized by different degree of roughness, $S_{q}$. It is found that the thermal contact resistance depends on the graphene loading, ξ, non-monotonically, achieving its minimum at the loading fraction of ξ ~15 wt%. Decreasing the surface roughness by $S_{q}\sim 1\;\mu m$ results in approximately the factor of ×2 decrease in the thermal contact resistance for this graphene loading. The obtained dependences of the thermal conductivity, $K_{TIM}$, thermal contact resistance, $R_{C}$, and the total thermal resistance of the thermal interface material layer on ξ and $S_{q}$ can be utilized for optimization of the loading fraction of graphene for specific materials and roughness of the connecting surfaces. Our results are important for the thermal management of high-power-density electronics implemented with diamond and other wide-band-gap semiconductors.

## 1. Introduction

A continuing trend of the miniaturization of electronic devices for information processing [1,2,3,4], and the increasing power density in high-power electronics [5,6,7,8,9,10] dictate the need for more efficient thermal management [11]. The reliability of devices and systems depend on their operating temperature [12]. Increasing device temperature results in an exponential increase in the rate of device failure [5,13]. Thermal interface materials (TIMs) are applied between the device and the heat spreader or heat sink to facilitate the heat transfer from the heat source to the environment (see Figure 1). Typically, less than 2% of the overall area interacts with each other when two surfaces, metallic or semiconductor, are placed in contact [14,15]. The remaining area is occupied by air, which has a low thermal conductivity of 0.026 Wm^−1^ K^−1^ at room temperature (RT) [16]. Filling the air gaps with TIMs that have substantially larger thermal conductivity comparing to that of the air is the main strategy for conventional thermal management approaches. Development of more efficient TIMs that can provide smaller thermal resistance for heat escape has become an important goal for electronic industry, and particularly for its segment, which deals with the high-power devices and systems [5,14,17].

The efficiency of the TIM connecting two surfaces is define by the total thermal resistance [18,19,20]:(1)$$R_{tot}=BLT/K_{TIM}+R_{C1}+R_{C2}.$$

Here, $K_{TIM}$ is the thermal conductivity of the TIM, $R_{C1}$ and $R_{C2}$ are the thermal resistances of the TIM layer with the two contact surfaces, and BLT is the bond line thickness, which is the thickness of the TIM layer. The $BLT/K_{TIM}$ represents the thermal resistance of the TIM layer. If TIM is used with the two identical surfaces, then $R_{C1}=R_{C2}=R_{C}$ and Equation (1) is simplified to:(2)$$R_{tot}=BLT/K_{TIM}+2R_{C}.$$

Minimizing BLT and $R_{C}$ reduces the overall thermal resistance, $R_{tot}$. These parameters depend on the thermophysical properties of the interlayer TIM and the roughness of the surfaces in contact. Roughness is determined by the nanoscale and microscale variations in the height profile of the physical surface. Typically, in modern electronics, BLT is assumed to vary from 25 to 100 μm [5,21]. For the thermal management of high-power-density electronics one may need larger *BLT* owing to the possible increase of roughness of the surfaces. For example, polycrystalline diamond, which can be used either as a substrate or active device layers is often characterized by large roughness due to the grains [22,23]. While many reports on new TIMs focus on the increase of the thermal conductivity, $K_{TIM}$, of the TIM composite, one should note from the above equation for $R_{tot}$, that the improvement of thermal management requires that TIM interfaces well with given surfaces, resulting in smaller $R_{C}$, and that BLT is optimized for a given surface roughness.

The main strategy for improvement of TIMs is incorporating thermally conductive fillers into the base polymer matrix, which can increase the overall thermal conductivity, $K_{TIM}$, of the TIM composites without substantially increasing $R_{C}$. In recent years, graphene has revealed its potential as a filler for both curing TIMs, e.g., with epoxy base, and noncuring TIMs, e.g., with silicone or other mineral oil bases. Graphene has extremely high intrinsic thermal conductivity, exceeding that of bulk graphite, which is $\sim 2000{\;\mathrm{Wm}}^{-1}K^{-1}$ near RT [24,25,26]. It was also established that few-layer graphene (FLG) maintains high thermal conductivity, similar to bulk graphite owing to its smooth surface and, as a result, insignificant reduction in thermal conductivity due to the phonon—boundary scattering [27,28,29,30]. A mixture of single-layer graphene and FLG demonstrated the largest enhancement in the thermal conductivity of the TIM composites [19,20,31,32,33,34,35,36,37,38,39,40,41,42,43,44,45,46,47,48,49]. In the context of thermal research and TIMs, we will refer to the processed mixture of graphene and FLG flakes with lateral dimensions in several μm range as *graphene fillers*. The thickness of the FLG fillers should be in the nanometer-scale range to preserve their flexibility. Such fillers can be produced inexpensively on a large industrial scale. The latter makes graphene TIMs much more practical than any composites with carbon nanotubes or other expensive materials.

Most of prior works on graphene TIMs report the thermal conductivity values of the composites and, in some cases, temperature rise experiments with specific device structures [34,50,51,52]. The questions of the thermal contact resistance of graphene TIMs with the surfaces of interest and the effects of roughness on the TIM performance have not been properly addressed. These are important issues for minimizing $R_{tot}$ for different electronic applications, particularly for the high-power density electronics where the surfaces can be characterized by larger roughness and hence, higher $R_{C}$. Here, we investigate the thermal contact resistance of the noncuring graphene TIMs with the surfaces characterized by different degree of roughness. The dependence of the total thermal resistance of the noncuring graphene TIMs on BLT is also obtained. In Section 2, we present experimental procedures. The discussion and conclusions are given in Section 3 and Section 4, respectively.

## 2. Synthesized Samples and Experimental Procedures

For this study, we used noncuring silicone—oil based TIMs with graphene and FLG fillers prepared from the commercial graphene powder (xGNP H-25, XG Sciences, Lansing, MI, USA, NAM). The noncuring graphene TIMs were applied to copper square plates (Midwest Steel Supply, Rogers, MN, USA, NAM) of thickness 1.09 mm and dimensions of 1 in × 1 in. The copper plates were polished (Allied High-Tech Products, Inc., Compton, CA, USA, NAM) and then treated with the sand paper to a different degree of roughness. A 3D optical profilometer (Profilm 3D, Filmetrics Inc., San Diego, CA, USA, NAM) was used to determine quantitatively the surface roughness values of the copper plates. The optical profiler utilized in this work operates on the basis of the non-contact optical technique of the white-light interferometry (WLI) plates [53]. The details of the preparation of noncuring graphene TIMs and surface treatment of the copper plates are described in the Methods Section. Figure 2 shows the results of the profilometer measurements for a set of copper plates. Figure 2a is the roughness of reference copper plate as received from the vendor. The sample was not polished by the polisher. To increase the surface roughness of the copper plates shown in Figure 2b–d, the plates were polished at 100 RPM for ~1, ~2.5 and ~3.5 min, respectively. The areal root mean square (RMS) roughness, $S_{q}$, determined for these plates was 0.05, 1.2, 2.5 and 3.1 µm, respectively. The preparation of the surfaces and the profilometer measurements allowed us to investigate the effect of roughness on thermal contact resistance with graphene TIMs.

Bulk thermal conductivity, total thermal resistance, $R_{tot}$, and thermal contact resistance of the TIM with the surface of the plates, $R_{C}$, were measured following the ASTM D5470-06 standard with an industrial TIM tester (LongWin Science and Technology Corp, Taiwan). The schematic of the measurement setup is shown in Appendix A. The TIM tester utilizes the steady-state method [54]. The measurement setup is comprised of two very flat steel plates with roughness in the range of a few nm as the heat source and sink. The TIM is applied between these plates. The heat flow and the temperature of the source and sink are carefully controlled. The thermal conductivity of TIM is extracted using the one-dimensional Fourier heat transport equation for given BLT of TIM. The details of the thermal testing are provided in the Methods Section. The initial measurements were performed on TIMs with different loading of graphene content without the copper plates. A layer of the synthesized TIM was applied between the two plates of the TIM tester. The BLT was controlled using the plastic shims. Note that the shims occupy a negligible portion of the area and volume of the TIM material that their contribution to overall heat transfer is negligible. All measurements have been performed under 0.55 MPa (~80 psi) of applied pressure, P.

## 3. Results and Discussion

We first measured the thermal properties of the prepared non-cured graphene TIMs without the copper plates. Figure 3a,b shows the total thermal resistance of graphene TIM, $R_{tot}$, as a function of BLT for different graphene loading, ξ. Figure 3a includes the thermal resistance of the silicone oil base as a reference. Figure 3b shows the data for the graphene loading of 10 wt% and more so that the trends can be seen more clearly. The total thermal resistance increases linearly with BLT as expected [55,56]. The data were used to plot a linear regression fitting for each loading fraction. For each fitting, the inverse of the line slope determines the bulk thermal conductivity of the TIM itself. The *y*-intercept of the fitted line is equal to the total thermal contact resistance, $2R_{C}$, of each TIM with the upper and lower contact surfaces. A table showing the obtained values is provided in the Appendix A).

Figure 4 shows the thermal conductivity of the noncuring graphene TIMs as a function of graphene loading, ξ. The thermal conductivity of the silicon oil base is $0.18\;{\mathrm{Wm}}^{-1}K^{-1}$. The thermal conductivity starts to increase rapidly with the addition of graphene. The increase is super-linear suggesting that the fillers form a percolated network facilitating the heat conduction. Note that in this Figure the *y*-axis is in logarithmic scale. At the loading of ξ=10 wt%, the increase in thermal conductivity slows down. This trend is consistent with prior studies for noncuring graphene composites [41], and different from that observed in curing epoxy composites with graphene [19,20,36,37,38,39,40,42]. In the cured solid TIMs, the thermal conductivity reveals linear to super-linear dependence on the filler loading [39]. The non-curing TIMs, on the other hand, exhibit a saturation effect for the thermal conductivity starting at some critical filler loading. This is similar to the effect reported previously for nano-fluids and soft TIMs [57,58,59,60]. The saturation effect can be explained by the tradeoff between the enhancement trend in the thermal conductivity as more fillers are added to the matrix and the decrease in the thermal conductance as the thermal interface resistance between the filler‒filler and filler‒matrix interfaces increases due to the incorporation of more fillers into the matrix [41]. In our noncuring TIMs, we achieved the value of the thermal conductivity of $\sim 4.2{\;\mathrm{Wm}}^{-1}K^{-1}$ at the graphene loading of 40 wt%. We intentionally did not increase the loading further due to the onset of the agglomeration. For the purpose of this study, it was important to have the consistent dispersion of the fillers. Overall, the thermal conductivity of graphene TIMs increased by the factor of ~19× for 30 wt% and 24× for 40 wt% loadings compared to the thermal conductivity of the silicone oil base.

In Figure 5, we present the measured thermal contact resistance of the TIM, $R_{C}$, as a function of graphene loading, ξ. $R_{C}$ is obtained by dividing the *y*-intercept of the fitted lines in Figure 3 by two (see Equation (2)). The measured $R_{C}\left(\xi \right)$ dependence revealed a rather unexpected non-monotonic trend. Contrary to the expectation of increasing $R_{C}$ with higher filler loading, we observe a rapid decrease in $R_{C}$ values up to the loading ξ=15 wt%, followed by a slow increase at the higher loading fraction. Theoretically, $R_{C}$ depends on the bulk thermal conductivity and shear modulus of the TIM and the roughness of the adjoining surfaces and the applied pressure. There is a trade-off between the thermal conductivity and shear modulus effect on $R_{C}$. The higher is the thermal conductivity, the lower is $R_{C}$, whereas for the shear modulus the dependence is vice versa [61]. Typically, one would want to increase the loading to improve $K_{TIM}\;$ as long as the viscosity and the shear modulus requirements allow for it. Based on the measured $R_{C}\left(\xi \right)$ dependence, one may prefer to limit the loading to smaller fraction in order to minimize $R_{tot}$. One should also note that increasing ξ limits the minimum attainable BLT.

Assuming that the “bulk” thermal conductivity of the TIM layer in semi-solid or semi-liquid TIMs is much smaller than that of the binding surfaces, the contact resistance can be described using the semi-empirical model as [57,61,62]:(3)$$R_{C1+C2}^{}=2R_{C}^{}=c\left(\frac{S_{q}}{K_{TIM}}\right){\left(\frac{G}{P}\right)}^{n},$$
where $G=\sqrt{{G^{\prime }}^{2}+{G^{\prime \prime }}^{2}}$. Here, $G^{\prime }$ and $G^{\prime \prime }$ are considered to be the storage modulus and the loss shear modulus of TIM, *P* is the applied pressure, $S_{q}$ is the average roughness of the two binding surfaces, and c and n are empirical coefficients, respectively. The two parameters of $K_{TIM}$ and G have opposite effects on the contact resistance, $R_{C}$, at a constant applied pressure. Increasing the graphene filler loading would result in an increase in both $K_{TIM}$ and G of the TIM layer. The equation also suggests that for TIMs with a specific filler, there exists an optimum filler loading where the thermal conductivity, $K_{TIM}$, would increase significantly while slightly effecting the thermal contact resistance. Combining Equations (2) and (3), one can write the total thermal resistance as:(4)$$R_{tot}^{}=\left(\frac{1}{K_{TIM}}\right)\left\{BLT+cS_{q}{\left(\frac{G}{P}\right)}^{n}\right\}.$$

In this form, the equation indicates clearly that an increase in the TIM thermal conductivity, $K_{TIM}$, can results in a reduction of the total thermal resistance.

To investigate the effect of the surface roughness on the thermal contact resistance with graphene TIMs, we measured the total thermal resistance, $R_{tot}$, of specially prepared copper plates with varying degree of roughness, $S_{q}$, using TIM tester (see Figure 2 and the Section 4.1). The samples were placed between the TIM tester’s heat sink and source which are made of very flat steel plates. A fraction of a droplet of silicone oil was added between the top and bottom copper plates with the heat sink and source to minimize the contact resistance between the copper and steel solid-solid interfaces. Note that in this case, the total thermal resistance, assuming a one-dimensional heat transport would be:(5)$$R_{tot}=BLT/K_{TIM}+2(R_{C,St-oil}+R_{C,oil-Cu}+L_{oil}/K_{oil}+L_{Cu}/K_{Cu}+R_{C,TIM-Cu}).$$

In this equation, $R_{C}$ is the thermal contact resistance between various interfaces defined by the subscripts. L and K are the thickness and bulk thermal conductivity of different components. The subscripts “St”, “Cu”, “oil”, and TIM, represent the steel, i.e., the heat source and sink of the TIM tester, copper plates, silicone oil, and TIM layer, respectively. We used TIMs with ξ=15 wt% and ξ=30 wt% to study the effects of roughness. We selected these two filler concentrations since at ξ =15 wt% the minimum in $R_{C}$ is attained whereas ξ=30 wt% provides a trade-off between the contact resistance and thermal conductivity—somewhat larger $R_{C}$ (see Figure 5) but enhanced $K_{TIM}$ as well (see Figure 4).

In Figure 6, we present the results of the total thermal resistance, $R_{tot}$, of noncuring graphene TIMs dispersed between two copper plates as a function of TIM’s BLT for two different graphene loadings, ξ, and four different values of roughness, $S_{q}$. For all the roughness values of copper plates and filler loadings, $R_{tot}$ increases with increasing BLT, as expected. This means that TIMs were dispersed properly without leaving unfilled air gaps. An interesting observation is that in some cases, the proper selection of BLT and graphene loading, ξ, can compensate for substantial increase in the roughness, $S_{q}$. Consider the case of TIM with ξ=30 wt% of graphene fillers and two roughness values $S_{q}=1.2\;\mu m$ (purple triangle symbols) and $S_{q}=3.1\;\mu m$ (violet hexagon symbols). The use of BLT~300 µm with the copper plates characterized by larger values of roughness, $S_{q}=3.1\;\mu m$, did not result in the overall increase in $R_{tot}$ as compared to the copper plates with $S_{q}=1.2\;\mu m$. The thermal resistance remained at $R_{tot}\;\sim \;2\;{\mathrm{Kcm}}^{2}W^{-1}$ (see Figure 6).

We extracted the *y*-intercept, b, of each data set presented in Figure 6 from the linear regression fittings, and related that to the total thermal contact resistance of the TIMs as a function of surface roughness. According to Equation (5), for sandwiched structures, the *y*-intercept of the plot is equal to:(6)$$b=2(R_{C,St-oil}+R_{C,oil-Cu}+L_{oil}/K_{oil}+L_{Cu}/K_{Cu}+R_{C,TIM-Cu}).\;$$

The thermal resistance of the copper plates is negligible ($2L_{Cu}/K_{Cu}\sim 7.3\times 10^{-4}\;{\mathrm{Kcm}}^{2}W^{-1}$). Therefore, the *y*-intercept of the graph in fact presents the summation of the total contact resistance of the sandwich structure, $R_{C,tot}=2(R_{C,St-oil}+R_{C,oil-Cu}+R_{C,TIM-Cu}$) plus the thermal resistance of the silicone oil layers at the copper-steel interfaces ($R_{oil}=2L_{oil}/K_{oil}$). In each measurement, the $R_{C,St-oil}$, $R_{C,oil-Cu}$, and $L_{oil}/K_{oil}$ are fixed values since the roughness of upper surfaces of all the copper plates at the interfaces with the heat source and sink and the thickness of the oil layer are the same. Therefore, the extracted values for the $R_{C,tot}+R_{oil}$ presented in Figure 7 indicate a measure for evaluating the contact resistance between the TIM layer and varying roughness of the copper plates. The determined values of $R_{C,tot}+R_{oil}$ as the function of the surface roughness, $S_{q}$, are shown in Figure 7 and listed in Appendix A. As seen, $R_{C,tot}+R_{oil}$ grows with the surface roughness which indicates an increase in $R_{C,tot}$ ($R_{oil}$ is a fixed value). The contact resistance for TIM with the higher loading, ξ=30 wt%, is larger than that with ξ=15 wt%. This is an expected trend for the oil-based noncuring TIMs as the loading fraction of fillers increases. The obtained results can help in optimization of TIM composition for applications with different surfaces, particularly those characterized by large roughness.

In high power electronic packaging, e.g., insulated gate bipolar transistors (IGBT) or silicon carbide-based device, a non-curing TIM layer is typically applied between the direct bond copper (DBC) layer and the heat sink [21,63,64,65]. This layer and its performance reliability [66,67,68] is usually the bottleneck of the packaging design since its thermal resistance is the highest among the other constituent components. Therefore, efforts have been focused on decreasing the thermal resistance of this layer by enhancing the bulk thermal conductivity of the TIM and reducing the BLT at the interface. By reducing the BLT layer, the effect of the contact resistance and roughness of the adjoining surfaces become more dominant. Recent endeavors towards application of diamond-based electronics improves the heat transport at device level owing to the high thermal conductivity of diamond. However, it still lacks proper treatment and dissipation of the generated heat at the system and packaging level where the high roughness of the diamond-based devices become problematic. Our results show that the change of roughness in the scale of ~1 µm substantially increases the thermal contact resistance by a factor of ×2 and hence, should be addressed properly in the packaging process. Our results also suggest that graphene-based TIMs with optimized filler loading can be a potential solution for high-power electronics owing to their improved thermal conductivity and low thermal contact resistance.

## 4. Conclusions

We investigated the thermal contact resistance of the noncuring graphene TIMs with the surfaces characterized by various degrees of roughness. It was found that the thermal contact resistance depends on the graphene loading non-monotonically, achieving its minimum at the loading fraction of ~15 wt% for the studied mixture of graphene fillers. Increasing the surface roughness by 1 μm results in the approximately factor of ×2 increase in the thermal contact resistance, $R_{C}$, for these TIMs. The total thermal resistance of the layer of the noncuring thermal interface material scales linearly with the bond-line thickness in the studies range from 5 to 35 µm. A projection to the micrometer bond-line thicknesses indicate that graphene thermal interface materials can meet the thermal management requirements for the high-power electronics. The obtained results are important for thermal management of high-power electronics implemented with diamond and other wide-band-gap semiconductors, which are typically characterized by a large degree of interface roughness.

### Methods

**Material synthesis:** Noncuring TIMs with graphene fillers were prepared from commercial FLG flakes (xGNP H-25, XG Sciences, Lansing, MI, USA, NAM) with the vendor specified average lateral dimension of ~25 µm. The mixture of graphene and FLG was weighed in a cylindrical container to obtain the desired filler concentration in each TIM sample. To maintain the quality and size of the fillers, acetone was added, thus ensuring that the fillers are not agglomerated during the mixing process [69]. The mixture of graphene—FLG fillers with acetone was introduced to silicone oil (Fisher Scientific, Hampton, NH, USA, NAM) base polymer, also known as PDMS—Poly(dimethylsiloxane). The weighing of each component was performed using the professional scale (Ohaus Corporation, Parsippany, NJ, USA, NAM). The resulting compound was then mixed using a high-speed shear mixer (Flacktek Inc., Landrum, SC, USA) at the speed setting of 300 rpm for 20 min. The role of the solvent, acetone, was to assist in obtaining the homogenous dispersion of the fillers in the base polymer. At the next step, acetone was evaporated in an oven (Across International, Livingston, NJ, USA, NAM) at ~70 °C for 2 h to ensure that it does not remain in the final TIM. The described method of TIM preparation is a modification of the procedure reported by some of us previously [41].

**Surface roughness:** For this study, we used copper square plates (Midwest Steel Supply, Rogers, MN, USA, NAM) of thickness 1.09 mm and dimensions of 1 × 1 in. The copper plates were polished to different degree of roughness using the Metprep 3 polisher (Allied High-Tech Products, Inc., Compton, CA, USA, NAM). The copper plates were polished with the 8-inch, 180 grit silicon carbide paper discs (Allied High-Tech Products, Inc). A 3D optical profilometer (Profilm 3D, Filmetrics Inc., San Diego, CA, USA, NAM) was used to quantitatively determine the surface roughness of the copper plates. The optical profiler operates on the basis of a non-contact optical technique of the white-light interferometry (WLI) [53]. A 50× Nikon Mirau interferometric objective lens was used to determine the surface profile of the plates. The *S_q_* roughness was defined as [70]:(7)$$S_{q}=\sqrt{\frac{1}{A}\iint_{\;}^{A}Z^{2}\left(x,y\right)dx\;dy}.$$

Here, *A* is the area and *Z(x,y)* is the surface profile amplitude. The 50× lens has been chosen to improve the accuracy of the data acquisition process.

**Thermal Characterization:** The thermal conductivity and thermal contact resistance of the TIMs were measured using the industrial TIM tester (LongWin Science and Technology Corp, Taiwan, Asia). The tester utilizes the steady-state method and meets the requirements of the industry standard ASTM D5470-06. The noncuring graphene TIMs were tested under the pressure of 80 psi and a temperature of 80 °C for 40 min for each thickness. The plastic shims (Precision Brand Products Inc., Downers Grove, IL, USA, NAM) were used to measure the graphene TIMs at different bond line thicknesses (BLT). The temperature at the heat source was constant for all measurements of TIMs at all measured BLT. TIMs were placed and tested in between the heat source and heat sink and is subjected to a temperature gradient at a set uniform pressure load. The pressure and temperature were constant to measure the thermal properties of TIMs. The measurements allowed us to determine the thermal conductivity and thermal contact resistance of each TIM [54]. The thermal properties of the TIMs were measured with the steel plates provided with the TIM tester. To determine the effect of surface roughness on the thermal contact resistance of TIMs, the instrument was calibrated for proper thickness using the copper plates with the same roughness. To avoid the presence of air gaps between the plates of the TIM tester and the copper plates an ultra-thin layer of silicone oil was used on each side. This process was consistently repeated for all sets of copper plates. The TIMs were then sandwiched between the copper plates. The plastic shims ensured the desired BLT. The constant test conditions were maintained during the testing of the TIMs with and without the copper plates. Additional details are provided in the Appendix A.

## Figures and Tables

**Figure 1 nanomaterials-11-01699-f001:**
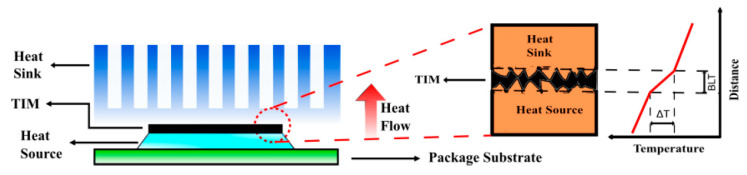
Schematic representation of a packaged device illustrating the role of TIMs. TIM is applied between the adjoining heat source and the heat sink surfaces in order to fill the air gaps and facilitate the heat transfer.

**Figure 2 nanomaterials-11-01699-f002:**
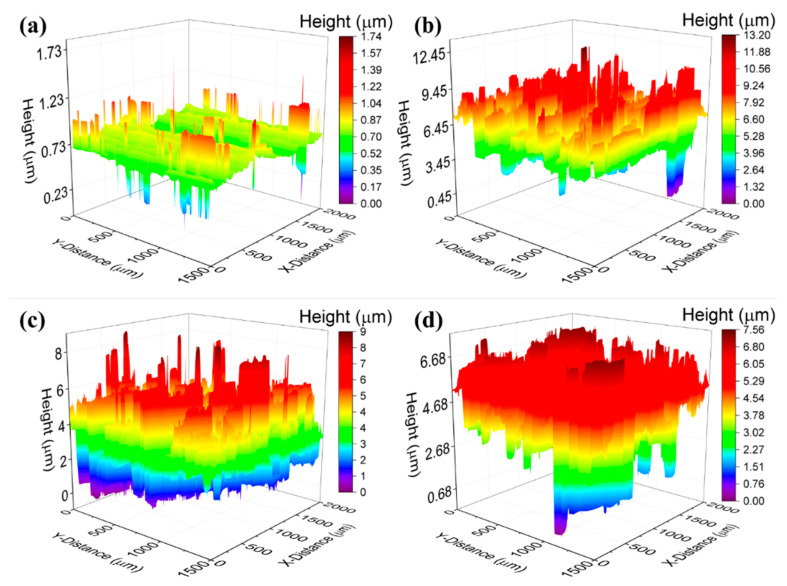
Roughness characteristics of the copper plates determined by an optical profilometer. The plates have the following root mean square (RMS) roughness: (**a**) $S_{q}$ = 0.05 µm, (**b**) $S_{q}$ = 1.2 µm, (**c**) $S_{q}$ = 2.5 µm, and (**d**) $S_{q}$ = 3.1 µm.

**Figure 3 nanomaterials-11-01699-f003:**
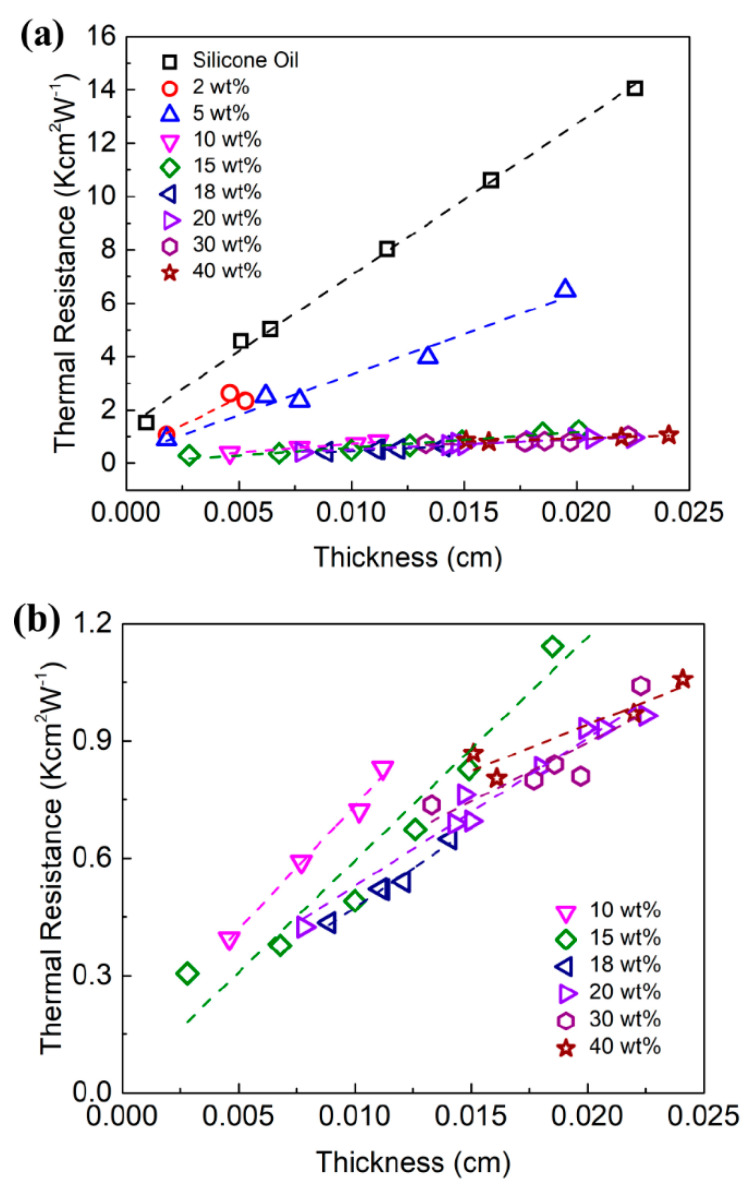
Total thermal resistance, $R_{tot}$, of graphene TIMs as a function of the bond line thickness, BLT. The dashed lines show the linear regression to the experimental data used to extract the thermal conductivity, $K_{TIM}$, and thermal contact resistance, $R_{C}$. (**a**) Thermal resistance vs. BLT for all tested loading fractions of graphene and pure silicone oil base. (**b**) Thermal resistance vs. BLT for loading fractions of ξ=10 wt% and above are shown for clarity.

**Figure 4 nanomaterials-11-01699-f004:**
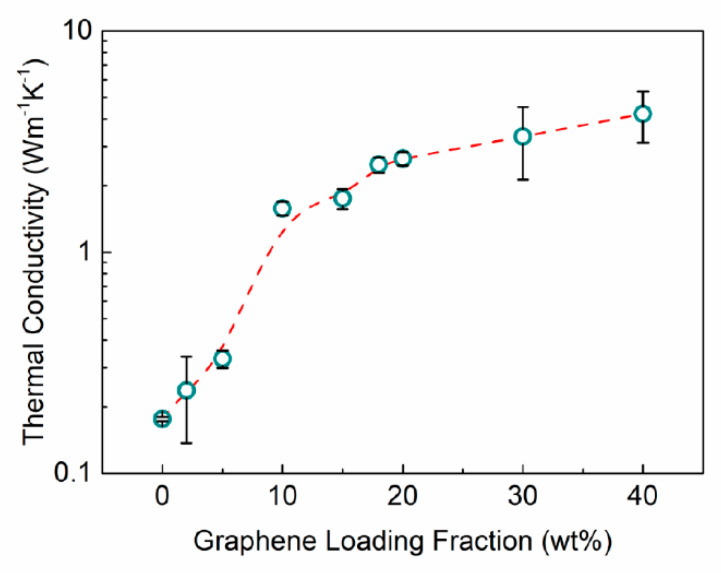
Thermal conductivity, $K_{TIM}$, of TIMs as a function of graphene loading fraction, ξ. Adding graphene fillers to the noncuring oil base increases the “bulk” thermal conductivity of graphene TIMs. The bars show the standard error of the linear regression slope.

**Figure 5 nanomaterials-11-01699-f005:**
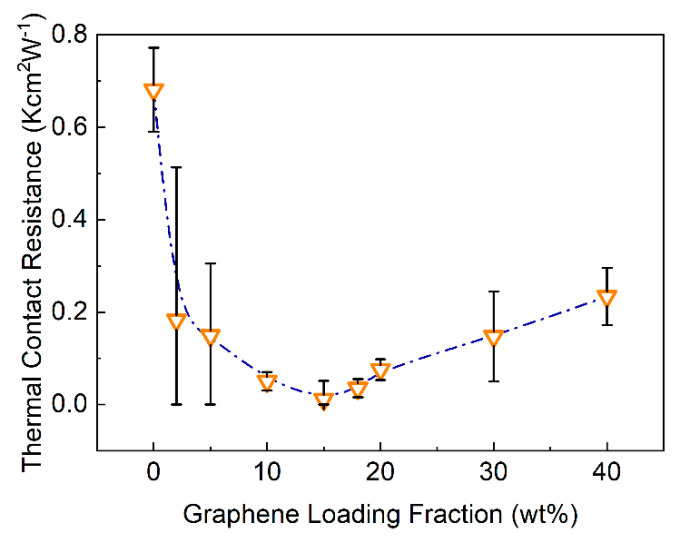
Thermal contact resistance, $R_{C}$, of TIMs as a function of the graphene loading, ξ. Note the non-monotonic dependence of $R_{C}$ on graphene loading. The bars show the standard error of the linear fittings used for data extraction.

**Figure 6 nanomaterials-11-01699-f006:**
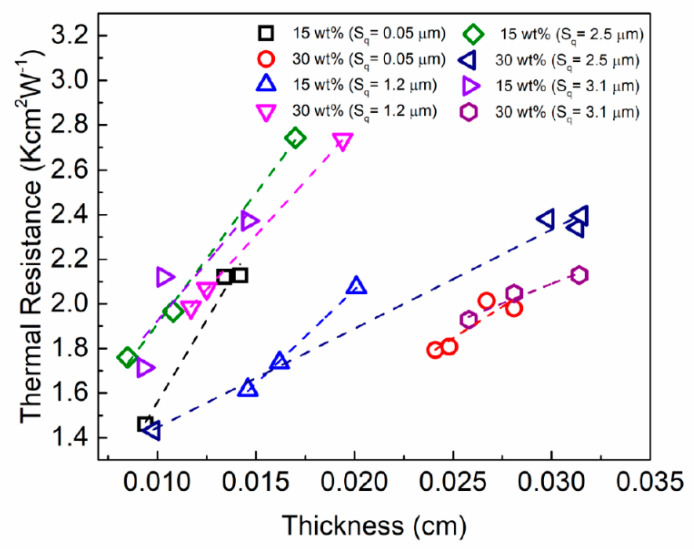
Thermal resistance, $R_{tot}$, of the graphene TIM dispersed between two copper plates as a function of the bond line thickness, BLT. The results are presented for two different graphene loadings, ξ, and four different values of roughness, $S_{q}$. In each measurement, the two copper plates used had the same roughness. The dashed lines show the linear regression fittings to the experimental data.

**Figure 7 nanomaterials-11-01699-f007:**
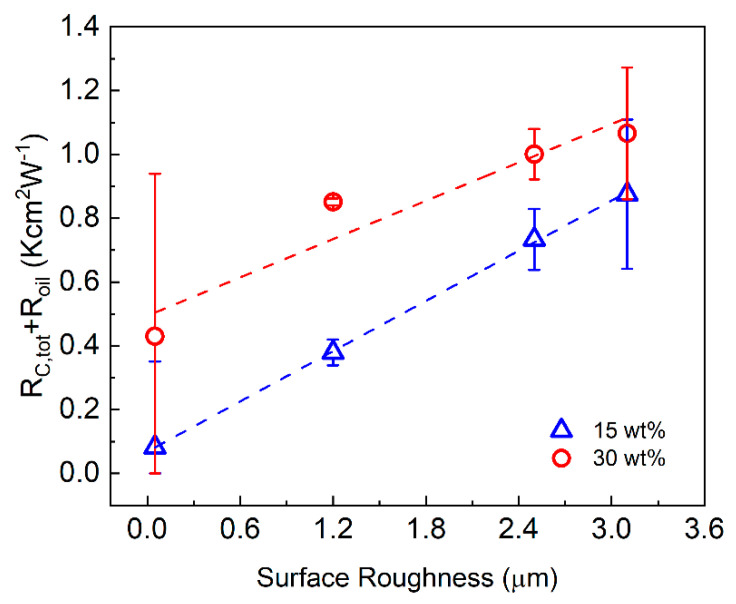
$R_{C,tot}+R_{oil}$, of graphene TIMs between two copper surfaces as a function of the surface roughness, $S_{q}$.

## Data Availability

Not Applicable.

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
