# Peer review of "Noncured Graphene Thermal Interface Materials for High-Power Electronics: Minimizing the Thermal Contact Resistance"

_nanomaterials, 2021, doi:10.3390/nano11071699_

Round 1

Reviewer 1 Report

Title: Noncured Graphene Thermal Interface Materials for High-Power Electronics: Minimizing the Thermal Contact Resistance

This manuscript reports experimental investigation of thermal contact resistance and thermal conductivity of noncuring graphene TIMs dependence on the surface roughness, graphene loading fraction and bond line thickness. This thermal contact resistance was found to have a non-monotonical dependence on the graphene loading. While it changed monotonically with the bond line thickness and surface roughness.  

This is an interesting topic by solving the questions of thermal contact resistance of graphene TIMs with the surface and effect of roughness on the TIM performance. These are important issues for the real high-power density electronic applications. 

However, the reported study lacks considerably in thoroughness, as outline below. I therefore would recommend this manuscript for publication in Nanomaterials when the following questions are addressed.

The central problems are the thermal gradient in the sample space and optimization for the surface roughness. For the thermal characterization part, the TIMs were sandwiched between copper plates (heat sink and heat source). The authors should explain during the thermal measurements which quantity remains unchanged? Is the thermal gradient in TIMs or the temperature difference between sink and source? If the temperature difference between sink and source remains constant, when the bond line thickness is varied in the experiment, the thermal gradient inside graphene TIMs will also change. In this case, the change in the thermal gradient will definitely lead to the change of thermal resistance. Then Figure 3 cannot be used to explain the thermal resistance dependence on the bond line thickness, other factors like thermal gradient should be also discussed. The second question is about the surface roughness, where it is induced by rotating the cooper plates. The title of this paper is ‘minimizing the thermal contact resistance’. To focus more on the title, the authors should explain more about how to reduce the surface roughness. Since currently the minimum surface roughness of 0.05 micrometer is the one not polished (line 93 and Figure 7).     

Followings are additional critical comments that may affect the conclusion of the manuscript: 

1, In the title and main text, the authors mentioned ‘high-power electronics’ many times, please give some real application examples of graphene TIMs in high-power electronics. 

2, I would suggest the authors to rewrite the abstract in a way to show how the thermal contact resistance can be minimized, instead of saying how the thermal contact resistance can be increase, e.g. ’increasing the surface roughness….’(line 13-14).

3, How the heat was transferred between sink and source in Figure 1, any temperature gradient? Did the heat transfer procedure influence the thermal contact resistance? 

4, In line 51, authors mention bond line thickness between 25 to 100 micrometers is preferred. Please explain why this thickness range is preferred and whether this range varies for different materials.

5, In line 89, authors mentioned 3D optical profile meter was used to determine the surface roughness. What is the error for the values? Besides, if the plates were polished with same rotation speed and time, will the surface roughness always stay same even for different samples? By what range, the surface roughness will change for the same polishing procedure? 

6, In line 116, the authors mentioned an applied pressure. Please explain whether the surface roughness changes the pressure applied to graphene TIMs.

7, In Figure 3, please illustrate the accuracy for the bond line thickness (what is the error for the thickness value combined with accuracy in 3D optical profile) since this graphene TIMs have a surface roughness. Please explain more about maintaining the same bond line thickness while varying the surface roughness.

8, In line 140, please explain why cured solid TIMs show a linear dependence on the filter loading. 

9, In Figure 4, what is the bond line thickness? Is it a constant value? 

Reviewer 2 Report

Thank you for the work. 

Data is nice. It will be good for the draft if the experimental part and the result and discussion part will be arranged nicely. in the current form its looks confusing and mixed. 

Other things are fine.

Thank you

Reviewer 3 Report

In this manuscript entitled ‘Noncured Graphene Thermal Interface Materials for High-Power Electronics: Minimizing the Thermal Contact Resistance’, the authors presented the results of their work on thermal contact resistance of the non-curing graphene thermal interface materials of different roughness. Overall, the manuscript provides a lot of technical results but, after reading the manuscript it does not leave the reader the impression that the problem stated in the introduction section has been properly addressed. The manuscript can be greatly improved if the authors work on the following comments:

  • Please change the format of all equations presented in the text. Merging equations in the main text makes it difficult for the reader to follow.
  • Experimental procedures and methods are effectively the same. The separation of both sections seems redundant.
  • Figure 2 axis labels are too difficult to visualize, please adjust.
  • Can the authors comment on the colour scheme of the plots in figure 2; what is the significance of red, yellow, and blue? If the colour scheme has a physical meaning, please provide scale bars to identify the meaning
  • What are the error bars for the results presented in figure 3?
  • Line 124 in the main text: “ The total thermal resistance increases linearly with BLT as expect. The data were used to plot….”. It is not evident to the reader why this result is expected at the first glance of this presented data. Can the authors comment on why with increasing graphene load, the thermal resistance should increase?
  • Is the linear regression fitting in figure 3 based on theoretical predictions or purely based on statistical analysis? If based on theory (new or existing) please comment on key equations needed to understand the results. Extrapolated results from figure 3 should be presented in the main text in a table.
  • Please further comment on the meaning of “super-linear” in line 137 of the main text and the relation to the percolation network.  
  • Please justify the equation presented in line 175 with either a derivation or an appropriate source to the origin of the equation.
  • Please comment on why the error bars are significantly higher for low graphene load fractions with respect to other fractions used.
  • Interpretation of results for figure 5 is based on the 1D heat transport equation for various stacks of material. Authors should provide simulation results of all the assumed equation/ theories indicated in the main text and compare with measured results. E.g. in line 188 the equation for the thermal resistance of 1D thermal transport is presented. Please provide simulation results of the 1D heat transfer equation(s) with a known parameter of the material and compare theoretical and experimental findings.
  • Insufficient data for figure 6. More data is needed for a variety of thicknesses to demonstrate the claimed trends.

Round 2

Reviewer 3 Report

The authors have addressed the concerns of the review. No further questions asked.